# Marine-Derived Natural Compounds for the Treatment of Parkinson’s Disease

**DOI:** 10.3390/md17040221

**Published:** 2019-04-11

**Authors:** Chunhui Huang, Zaijun Zhang, Wei Cui

**Affiliations:** 1Ningbo Key Laboratory of Behavioral Neuroscience, Zhejiang Provincial Key Laboratory of Pathophysiology, School of Medicine, Ningbo University, Ningbo 315211, China; 156001359@nbu.edu.cn; 2Laboratory of Marine Natural Products, School of Marine Sciences, Ningbo University, Ningbo 315211, China; 3Institute of New Drug Research, Guangdong Province Key Laboratory of Pharmacodynamic, Constituents of Traditional Chinese Medicine and New Drug Research, College of Pharmacy, Jinan University, Guangzhou 510632, China; zaijunzhang@163.com

**Keywords:** marine-derived natural compounds, Parkinson’s disease, α-synuclein, oxidative stress, neuroprotective, clinical trials

## Abstract

Parkinson’s disease (PD) is a neurodegenerative disorder caused by the loss of dopaminergic neurons, leading to the motor dysfunctions of patients. Although the etiology of PD is still unclear, the death of dopaminergic neurons during PD progress was revealed to be associated with the abnormal aggregation of α-synuclein, the elevation of oxidative stress, the dysfunction of mitochondrial functions, and the increase of neuroinflammation. However, current anti-PD therapies could only produce symptom-relieving effects, because they could not provide neuroprotective effects, stop or delay the degeneration of dopaminergic neurons. Marine-derived natural compounds, with their novel chemical structures and unique biological activities, may provide anti-PD neuroprotective effects. In this study, we have summarized anti-PD marine-derived natural products which have shown pharmacological activities by acting on various PD targets, such as α-synuclein, monoamine oxidase B, and reactive oxygen species. Moreover, marine-derived natural compounds currently evaluated in the clinical trials for the treatment of PD are also discussed.

## 1. Introduction

Parkinson’s disease (PD) is a common neurodegenerative disease that mainly occurs in the elderly [1,2]. The main pathological changes of PD included the progressive loss of dopaminergic (DA) neurons in the substantia nigra pars compacta (SNc), the decrease of dopamine content in the striatum, and the formation of α-synuclein aggregates in the brain [3]. The death of DA neurons in SNc could further inhibit thalamic activity, and reduce the excitatory input to the motor cortex, resulting in slow movements and limbs stiff in PD patients [4].

Although the etiology of PD is still unclear, the death of DA neurons during PD progress was revealed to be associated with the abnormal aggregation of α-synuclein, the elevation of oxidative stress, the decrease of mitochondrial functions, and the increase of neuroinflammation [5]. Anti-PD drugs mainly aim to relieve motor and non-motor symptoms, and produce neuroprotective effects [6]. Clinically used anti-PD drugs include dopamine precursors (levodopa and carbidopa), dopamine agonists (pramipexole, ropinirole, rotigotine and apomorphine), catechol-O-methyl transferase (COMT) inhibitors and monoamine oxidase B (MAO-B) inhibitors (selegiline, rasagiline and safinamid) [7]. Levodopa is the first-line anti-PD drug used for relieving motor symptoms. As adjuvant therapies to levodopa, dopamine agonists are usually used in young PD patients. COMT inhibitors could increase the bioavailability of dopamine in the central nervous system (CNS) by reducing peripheral dopamine metabolism. MAO-B inhibitors can be used as a mono-therapy, or combined with t dopamine agonists or levodopa. However, these anti-PD drugs could relieve motor or non-motor symptoms, but could not produce neuroprotective effects. Therefore, the disease progress of PD could not be stopped or delayed by these drugs. Moreover, some anti-PD drugs could produce serious side effects. For example, levodopa could block aromatic amino acid decarboxylase, and cause nausea, vomiting, insomnia and nightmares. Long-term use of levodopa could lead to “on-off” phenomena, a switch between immobility and mobility in PD patients [8]. MAO-B inhibitors could increase synaptic availability of dopamine, and lead to nausea, insomnia, hallucinations and serotonergic crisis in combination with selective serotonin reuptake inhibitors [8]. Therefore, it is necessary to discover drugs that can effectively treat PD with neuroprotective effects and few side effects.

Marine-derived natural compounds could produce a variety of pharmacological effects, and are potentially highly useful for the development of new drugs [9]. 7 marine-derived natural compounds have been approved for clinical use [10]. Here, we have summarized marine-derived natural compounds with the effectiveness for the treatment of PD (Table 1). Moreover, marine compounds that have entered clinical trials against PD have also been discussed (Table 2).

## 2. The Therapeutic Targets of PD

Many targets, including the abnormal aggregation of α-synuclein, MAO-B, neurotrophic factors, and reactive oxygen species (ROS) have crucial impacts on the progress of PD.

### 2.1. α-Synuclein Aggregates

The aggregation of α-synuclein is associated with neurotoxicity in the brain, leading to the progressive loss of DA neurons during PD progress. Therefore, abnormal α-synuclein aggregation has been intensively considered as a main therapeutic target for PD [88]. Moreover, Lewy bodies, protein inclusion bodies contain mainly aggregated α-synuclein, are presented in the pre-synaptic neurons [89]. Accumulation of α-synuclein aggregation could be produced either by the excessive aggregation of α-synuclein or by the incapable clearance of aggregated α-synuclein [90]. Abnormal aggregates of α-synuclein, such as oligomers or fibrils, may interfere with cell processes, leading to the diffusion of α-synuclein aggregates between neurons [91]. It has been shown that wild-type mice treated with aggregates of α-synuclein in the striatum could cause the intracellular transmission of α-synuclein in the whole brain. Moreover, the administration of α-synuclein aggregates results in the loss of DA neurons in SNc, the decrease of dopamine levels in the striatum, and eventually the motor deficits in PD [92]. Although it is not clear how α-synuclein aggregates cause neuronal toxicity, there is evidence that α-synuclein aggregates may disrupt protein degradation systems and mitochondrial functions [93]. Therefore, inhibiting the expression and aggregation of α-synuclein or increasing the clearance rate of aggregated α-synuclein may be an important strategy for the treatment of PD [94].

### 2.2. MAO-B

MAO-B mainly located in the outer membrane of mitochondria [95]. Unlike monoamine oxidase-A (MAO-A), which is involved in the metabolism of serotonin, norepinephrine and dopamine, MAO-B is specialized to metabolize dopamine neurotransmitters, and produces ROS that directly damages DA neurons [96]. Therefore, MAO-B inhibitors could elevate the concentration of synaptic dopamine by blocking the degradation of dopamine, and eventually produce neuroprotective effects [97]. For example, selegiline, an irreversible MAO-B inhibitor, could produce neuroprotective effects *in vitro* [98]. Rasagiline, another MAO-B inhibitor, could be used to prevent the symptoms of idiopathic PD [99]. In animals, selegiline and rasagiline could prevent 1-methyl-4-phenyl-1,2,3,6-tetrahydropyridine (MPTP)-induced toxicity to DA neurons through the inhibition of neurotoxicity produced by 1-methyl-4-phenyl-pyridinium (MPP^+^), an MPTP metabolite [100]. Therefore, novel MAO-B inhibitors have been extensively studied for treating PD [101].

### 2.3. Neurotrophic Factors

Neurotrophic factors could promote the survival of neurons, and therefore be served as potential therapeutic agents for the treatment of PD [102]. It has been found that the glial cell-derived neurotrophic factor (GDNF) could enhance the differentiation and survival of DA neurons, producing anti-PD neuroprotective activity [103]. Moreover, other neurotrophic factors, such as neurturin, could protect and regenerate DA neurons [104]. Recent studies have shown that SKF38393, a D1 receptor selective agonist, could improve the growth of DA neurons by increasing the expression of brain-derived neurotrophic factor (BDNF) in neuronal cultures [105]. Moreover, natural compound PYM50028 could lead to a significant increase in GDNF and BDNF, reducing MPTP-induced loss of DA neurons in mice [106]. These studies support the idea that neurotrophic factors have therapeutic potential by preventing DA neuronal damage during PD progress, and drugs that could elevate the expression of neurotrophic factors might be used in the treatment of PD.

### 2.4. ROS

DA is not only a physiological neurotransmitter, but also a main source of ROS. DA contains an unstable catechol moiety, which can oxidize to form ROS [107,108]. In addition, the oxidation products of DA could be polymerized to form neuromelanin, a highly cytotoxic substance [109]. Many neurotoxins, such as MPTP, 6-hydroxydopamine (6-OHDA) and paraquat (PQ) could induce the elevation of ROS, and lead to similar pathological and biochemical symptoms as PD in humans [110]. Therefore, these neurotoxins are commonly used to establish PD animal models.

Many endogenous proteins could produce antioxidant activities. For example, DJ-1, a conserved protein involving in neurodegeneration, could produce neuroprotective effects against oxidative stress via activating the extracellular signal-regulated kinase (ERK) pathway [111]. Moreover, hydrogen peroxide (H_2_O_2_)-induced neurotoxicity is inhibited by a pro-survival phosphoinositide 3-kinase (PI3-K)/Akt pathway [112]. The transcription factor nuclear factor-erythroid 2-related factor 2 (Nrf2) could regulate the transcription of endogenous protective proteins, and protect against neuroinflammatory and oxidative stresses [113]. The Nrf2/antioxidant responsive element (ARE) cascade can be further involved in the maintenance of mitochondrial homeostasis [114]. Voltage-dependent anion-selective channel 1 (VDAC1) is located in the outer membrane of mitochondria. ROS produced by mitochondrial complex III could be released into the cytosol from VDAC1 [115]. In the brain of PD patients, VDAC1 could also increase the concentrations of Ca^2+^ in the mitochondria, leading to the increase of mitochondrial permeability, the disruption of mitochondrial membrane potential, and eventually to neuronal apoptosis and degeneration [116]. Therefore, neuroprotective agents that could elevate anti-oxidative stress via acting on VDAC1, ERK, PI3-K/Akt and Nrf2/ARE cascades might produce anti-PD activities.

To date, PD is a complex disease with multiple pathological factors. Therefore, the effects of multiple factors coordination in PD progress should not be negligible. Multi-target drugs could simultaneously act on two or more anti-PD targets, and might produce greater therapeutic potential than single-target drugs for treating PD. 

## 3. Potential Candidates from Marine-Derived Compounds for the Treatment of PD

### 3.1. Archaea

Many hyperthermophiles could release zwitterionic natural products in extreme environments such as high temperature and osmotic pressure to prevent thermal denaturation and aggregation of proteins [117]. Such natural products may be used to inhibit abnormal α-synuclein aggregation. Mannosylglycerate (MG) (**1**) is a compatible solute produced by hyperthermophilic bacteria in the hot environment (Figure 1). MG could inhibit the formation of α-synuclein aggregates in a PD yeast model. Moreover, MG could promote the folding of α-synuclein, preventing the pathological aggregation of α-synuclein [11]. Therefore, MG represents a good candidate for the treatment of PD.

### 3.2. Bacteria

Secondary metabolites from marine-derived bacteria represent a rich source for drug development with novel chemical structures and diverse biological activities [118,119]. NP7 (**2**) is a marine-derived compound from *Streptomyces* sp (Figure 2). NP7 is an antioxidant, and could pass the blood-brain barrier. NP7 at 5–10 μM could prevent apoptosis and necrosis induced by H_2_O_2_ in neurons and glial cells [12]. In addition, NP7 could inhibit microglial activation, and prevented the increased phosphorylation of ERK induced by H_2_O_2_. Therefore, NP7 may be developed as a neuroprotective agent against oxidative stress in PD [13,14].

The inhibitory activity of piloquinones, marine-derived compounds isolated from *Streptomyces* sp., on MAO-B was reported [120]. Piloquinone A (**3**) and piloquinone B (**4**) were isolated from *Streptomyces* sp. CNQ-027 (Figure 2) [15]. Piloquinone (A) is a potent inhibitor of MAO, with an IC_50_ value of 1.21 μM for MAO-B and an IC_50_ value of 6.47 μM for MAO-A, respectively. At the same condition, piloquinone (B) is only effective against MAO-B, with an IC_50_ value of 14.50 µM [16]. These results indicated that piloquinone derivatives may be useful lead compounds in the development of MAO-B inhibitors for the treatment of PD.

### 3.3. Fungi

There are many marine fungal metabolites could produce anti-PD neuroprotective activities. Neoechinulin A (**5**) is a diketopiperazine alkaloid of isoprene quinone isolated from the red algae-related fungus *Microsporum* sp. and *Aspergillus* sp. (Figure 3) [17]. Neoechinulin A could protect PC12 cells against MPP^+^- and peroxynitrite-induced neuronal death via reversing mitochondrial complex I dysfunction [18,19]. Moreover, Neoechinulin A could prevent rotenone-induced neurotoxicity by activating mitochondrial-related cytoprotective mechanisms [20]. Therefore, neoechinulin A may have the potential to interfere with progressive DA degeneration of PD.

Xyloketal B (**6**) belongs to a series of new ketal compounds isolated from the mangrove fungus *Xylaria* sp. (Figure 3) [21]. Xyloketal B could scavenge di(phenyl)-(2,4,6-trinitrophenyl)imino-nitrogen (DPPH) free radicals, and protect PC12 cells against ischemia-induced neurotoxicity [22]. Moreover, the neuroprotective effects of xyloketal B may be due to the inhibition of NADPH oxidase-derived ROS production [23]. However, the effects of xyloketal B on the prevention of MPP^+^-induced neurotoxicity are only moderate. Therefore, 39 new xyloketal derivatives were synthesized to screen drug candidates for the treatment of PD [121].

Secalonic acid A (**7**) is a natural product obtained from marine fungus *Aspergillus ochraceus* and *Paecilomyces* sp. (Figure 3) [24]. Secalonic acid A at 3–10 μM significantly inhibited colchicine-induced apoptosis of cortical neurons by the inhibition of JNK and p38 phosphorylation, and the reduction of Ca^2+^ influx and caspase-3 activation [25]. Secalonic acid A could also protect DA neurons against MPTP/ MPP^+^ by reversing mitochondrial apoptotic pathways [26].

Melatonin analog 6-hydroxy-*N*-acetyl-β-oxotryptamine (**8**), 3-methylorsellinic acid (**9**) and 8-methoxy-3,5-dimethylisochroman-6-ol (**10**) were isolated from marine fungus *Penicillium* sp (Figure 3). KMM 4672. These compounds could produce neuroprotective activity against 6-OHDA- and PQ-induced neuronal toxicity [27,28]. Similarly, candidusin A (**11**) and 4-dehydroxycandidusin A (**12**) from *Aspergillus* sp. KMM 4676, and diketopiperazine mactanamide (**13**) from *Aspergillus flocculosus*, can protect Neuro2A cells from the toxicity effects of 6-OHDA via scavenging ROS (Figure 3) [29,30,31]. These results suggested that such compounds may be used to develop anti-PD leads.

### 3.4. Algae

Marine algae are a rich source of antioxidants [122]. Carotenoids, particularly astaxanthin (**14**) derived from marine microorganisms (algae such as *Haematococcus pluvialis* and *Chlorella zofingiensis*), is proven to be an effective adjuvant therapy for preventing and/or delaying the progression of neurodegenerative diseases (Figure 4) [32,123]. The unique chemical structure of astaxanthin allowed it to easily cross the blood-brain barrier [33]. Moreover, astaxanthin could produce anti-PD effects in mice [34]. It was demonstrated that astaxanthin could reduce the activation of microglia in the brain of mice [33]. Astaxanthin could also inhibit neuronal apoptosis and mitochondrial abnormalities, and reduce intracellular ROS [35,36]. In addition, synthetic astaxanthin is superior to algae-extracted astaxanthin in terms of anti-inflammatory and anti-oxidant properties, leading to astaxanthin that is convenient and promising for the treatment of PD [37].

Algae polysaccharides are effective free radical scavengers and antioxidants *in vitro* [124]. Fucoidan (**15**), a representative algae polysaccharide, could prevent MPTP-induced neurotoxicity via its antioxidant and anti-apoptotic abilities [125]. Moreover, fucoidan could inhibit lipopolysaccharide (LPS)-induced nitric oxide (NO) production in microglia. The underlying mechanisms of these effects may include the down-regulation of intracellular ROS and pro-inflammatory cytokines [126,127]. 

Brown algae could produce a variety of fucoidan. The polysaccharide fucoidan is one of the main sulfated polysaccharides extracted from *Turbinaria decurrens* [38,124]. Mice treated with polysaccharide fucoidan could prevent MPTP-treated behavioral abnormities, possibly by increasing tyrosine hydroxylase (TH) levels in the substantia nigra and striatum [39]. These results indicated that polysaccharide fucoidan may produce neuroprotective effects against PD.

Sulfated hetero-polysaccharides (DF1) (**16**) and sulfated galactofucan polysaccharides (DF2) (**17**), another two fucoidan derived from brown seaweed *Laminaria japonica*, also possess neuroprotective activities [40]. The antioxidant activity of sulfated polysaccharides depends on their molecular weight, types of glucose and glycosidic branching [128]. DF1 and DF2 could significantly increase the number of DA neurons in MPTP-induced mice [41]. Moreover, DF1 and DF2 could reverse MPP^+^-induced decrease of mitochondrial activity [41]. DF1 has a more complex chemical composition, and shows the greater neuroprotective activity than DF2 [41]. In addition, DF1 could protect SH-SY5Y cells against H_2_O_2_-induced apoptosis by activating the nerve growth factor (NGF) /PI3-K/Akt signaling pathway [42]. 

*Spirulina platensis* (**18**) is a multicellular filamentous *cyanobacterium* with high levels of protein and a large amount of essential fatty acids [43]. *Spirulina platensis* protein extract and phycocyanin could exert antioxidant activity by elevating the activity of cellular antioxidant enzymes, indicating that *Spirulina platensis* is a powerful antioxidant that interferes with free radical-mediated cell death through mechanisms associated with antioxidant activity. The enhanced diet by *Spirulina platensis* could provide neuroprotection in α-synuclein-induced neurotoxicity model of PD [44]. Moreover, Spirulina polysaccharides could produce protective effects against MPTP- and 6-OHDA-induced loss of DA in mice [45,46].

Fucoxanthin (**19**) is a marine carotenoid extracted from edible brown algae, and was reported to have anti-oxidant and anti-inflammatory activity (Figure 5) [47]. Recently, fucoxanthin was found to produce neuroprective effects through multiple targets. Fucoxanthin could attenuate neurotoxicity induced by H_2_O_2_ via activating the PI3-K/Akt cascade and inhibiting the ERK pathway [48]. These reults suggested that fucoxanthin may play a role in the treatment of PD.

### 3.5. Sponge

Marine sponges are a rich source of bioactive chemicals [129,130]. Gracilins (A, H, K, J and L) (**20**) and tetrahydroaplysulphurin-1 (**21**) are a group of diterpene derivatives isolated from *Spongionella* sp., acting on tyrosine kinases (Figure 6) [49]. Both gracilins and tetrahydroaplysulphurin-1 could protect cortical neurons against oxidative damage by activating the Nrf2/ARE pathway, indicating that these metabolites may be interesting candidates for neurodegenerative diseases [50].

24-hydroperoxy-24-vinylcholesterol (**22**) is isolated from sponge *Xestospongia testudinaria* (Figure 6) [51]. 24-hydroperoxy-24-vinylcholesterol is an unusual oxidative steroid containing a hydroperoxy group in the branch. It has shown that the activation to nuclear factor kappa B (NF-κB) with an IC_50_ value at 31.3 μg/mL, suggesting that it might be useful for PD therapy [52].

Iotrochotazine A (**23**), contains a 1,1-dioxo-1,4-thiazine ring and a coumarin backbone, representing a novel structural class of marine alkaloid (Figure 6). Iotrochotazine A is extracted from *Iotrochota* sp. [53]. Iotrochotazine A could specifically affect the morphology and cellular distribution of lysosomes and early endosomes in olfactory neurosphere-derived cells from idiopathic PD patients, suggesting that iotrochotazine A may be developed as a novel anti-PD lead [54,55].

Mirabamides A-H peptide (**24**) is a group of marine peptides from the sponges *Siliquariaspongia mirabilis* and *Stelletta clavosa* [56,57]. It have been reported that elevated advanced glycation endproducts (AGEs) could induce the abnormal aggregation of α-synuclein [58]. Mirabamides A-H peptides could preferably inhibit pyridoxamine-based AGEs, and therefore might be developed as anti-PD candidates [59].

### 3.6. Coral

Marine-derived compound 11-dehydrosinulariolide (**25**) was extracted from soft coral *Sinularia flexibili* (Figure 7) [60]. 11-dehydrosinulariolide could up-regulate the PI3-K/Akt pathway to protect DA neurons against 6-OHDA-mediated damage [61]. Moreover, 11-dehydrosinulariolide-induced up-regulation of DJ-1 protein expression was revealed by 2D gel electrophoresis [62]. Furthermore, 11-dehydrosinulariolide could attenuate the damage of DA neurons in 6-OHDA-induced zebrafish and rats [63]. These results suggested that 11-dehydrosinulariolide can exert neuroprotective effects in PD models.

### 3.7. Mollusk

Staurosporine (AM-2282) (**26**) is a kinase inhibitor originally isolated from the actinomycete *Streptomyces staurosporeus* (Figure 8) [64]. Staurosporine (AM-2282) could also be found in marine sea squirt and flatworm [65]. By activating AMP-activated protein kinase (AMPK)/mammalian target of rapamycin (mTOR) signaling pathway, staurosporine (AM-2282) at 10 nM could induce DA neurite outgrowth in mesencephalic primary cultures [66]. Moreover, staurosporine (AM-2282) could protect neurons against ischemic-induced toxicity [67]. However, staurosporine is very toxic. Therefore, staurosporine analogues have been developed from structural modification with the aim of reducing toxicity [131].

### 3.8. Sea Cucumber

Sea cucumber is a marine invertebrate that contains valuable ingredients. In many Eastern counties, sea cucumber is recognized as a tonic and traditional therapy for neurodegenerative disorders. Whole body-ethyl acetate (WBEA), whole body-butanol (WBBU), and body wall-ethyl acetate (BWEA) (**27**) are extracts of sea cucumber *Holothuria scabra*. These compounds could significantly attenuate the degeneration of DA neurons induced by 6-OHDA in *caenorhabditis elegans* [68]. Moreover, these extracts could inhibit abnormal aggregation of α-synuclein, and restore lipid content [68].

The sea cucumber extract glucocerebrosides (SCG-1, SCG-2, and SCG-3) (**28**) from *Cucumaria frondosa* are important sphingolipids (Figure 9) [69]. SCG-1, SCG-2, and SCG-3 could promote neurite outgrowth in NGF-induced PC12 cells in a dose-dependent and structure-selective manners, possibly via enhancing NGF-induced TrkA phosphorylation and increasing BDNF expression [70]. These results suggested that sea cucumber extracts and its active ingredient compounds may have anti-PD potential.

### 3.9. Conus

Nicotine acetylcholine receptors (nAChRs) could regulate dopamine release, and are regarded as important target of PD [132]. Many marine organisms could produce toxins that selectively target nAChRs. α-conotoxin (TxIB) (**29**) derived from *Conus textile* is a 16 amino acid peptide (Figure 10). α-conotoxin could block α6/α3β2β3 nAChR with an IC_50_ value of 28 nM, indicating that α-conotoxin may be a potential candidate against PD [71].

## 4. Marine-Derived Drugs for Clinical Trials of PD

### 4.1. Omega-3 Fatty Acids (**30**)

The consumption of marine fish and seafood has been associated with mental health. Most of neurological benefits provided by seafood consumption is believed from an adequate uptake of omega-3 fatty acids (*n*-3/PUFAs) [133]. In the CNS, cell membranes of neurons contain (*n*-3)/PUFAs such as docosahexaenoic acid (DHA) and eicosapentaenoic acid (EPA) [72]. Omega-3 fatty acids are mainly contained in deep-sea oily fish, as well as algae [73]. Omega-3 fatty acids are used as a therapy in 1356 clinical trials, including 2 clinical trials related to PD. In a double-blind, randomized, placebo-controlled clinical trial, 29 PD patients receiving omega-3 fatty acid supplements completed a 12-week trial. The results showed that PD patients taking fish oil and antidepressants largely relieved depressive symptoms, suggesting that omega-3 fatty acid supplements are safe to PD patients, and may be used as an adjunct to other drugs [74]. In another double-blind, randomized, placebo-controlled clinical trial, after taking omega-3 fatty acids and vitamin E, 60 PD patients have beneficial effects at the unified PD rating stage [75].

### 4.2. Inosine (**31**)

Inosine, a precursor of uric acid, can be isolated from sponges (Figure 11) [76]. As an important physiological antioxidant, uric acid has been identified as a molecular predictor associated with a reduction in PD risk and a potential neuroprotective agent for PD [77]. In the clinical trial, inosine was generally safe, tolerable and effective in raising serum and cerebrospinal fluid urate levels in PD patients. These findings support further development of inosine as a potential disease-modifying therapy for PD [134].

### 4.3. Pramipexole (**32**)

Pramipexole is a dopamine agonist, and could be used in the treatment of PD (Figure 12). Marine yeast in deep seabed sediments can be used as a biocatalyst for stereo-selective reduction of different ketones and biotransformation in seawater, and could be used to produce pramipexole [78]. The NIH database showed that there are 79 clinical trials of pramipexoleon on PD. Particularly, improved depressive symptoms were observed in PD patients using pramipexoleon [79].

### 4.4. CEP-1347 (KT7515) (**33**)

CEP-1347, a semisynthetic indolocarbazole derivative isolated from naturally occurring *Nocardiopsis* sp. K252a (Figure 13) [80]. CEP-1347 could inhibit the SAPK/JNK pathway, which is activated after a variety of neuronal toxic insults in neurons [81]. Moreover, CEP-1347 is a potent inhibitor of mixed lineage kinase (MLK), and could be used to treat HIV-associated cognitive disorders by binding to ATP site of MLK [82]. However, CEP-1347 did not show effective therapeutic effects in early PD patients in a double-blinded, randomized, placebo-controlled clinical trial [83].

### 4.5. GM1 ganglioside (**34**)

GM1 ganglioside has been suggested as a disease-modifying treatment for PD (Figure 14) [84]. Monosialoganglioside GM1 can be prepared using a sialidase-producing marine bacterium as a microbial biocatalyst [85]. PD patients taking GM1 ganglioside showed early improvements of symptom and slow progression of symptoms. The results of the imaging studies provided additional data demonstrating the potential disease-modifying effects of GM1 on PD [86]. At another clinical trial, PD patients taking GM1 ganglioside showed a significant improvements in motor scores [87]. It is speculated that GM1 ganglioside may modulate lipid raft structure and function to exert neuroprotective activities. However, the detailed mechanism of neuroprotective effects of GM1 ganglioside is still uncertain.

## 5. Conclusions

Currently, there is no effective pharmacological treatment for blocking or slowing PD process. As aging continues to increase, the incidence of PD and explosive mortality of PD patients are increasing dramatically. Therefore, the discovery and development of new anti-PD drugs should not be delayed.

PD is a complex disease with many pathological factors. Therefore, it is possible to discover drugs that work synergistically with multiple targets of PD. Most clinically used anti-PD drugs are produced by acting on a single target with a certain degree of side effects. Marine-derived compounds usually contain reactive groups such as -OH, -NH_2_, and –SH in their chemical structures, and might be act as anti-oxidants. Other functional groups of marine-derived compounds might enable them to act on anti-PD targets such as α-synuclein, MAO-B and other key proteins in the signaling pathways. Therefore, marine-derived compounds, e.g., **7**, **14**, **16**, **17**, **18**, **25**, **26**, **27**, **28** in this review, might be developed as anti-PD leads with multiple targets.

Many clinical trials on PD aim to improve the efficacy of drugs already used in the treatment of PD. This is partially because of the difficulty of finding novel natural compounds from terrestrial organisms. The sea is a treasure for discovering novel natural compounds. Marine-derived compounds have continuously entered into clinical trials against different diseases, and have become one of the important sources of drug development [135]. Recently, the anti-AD marine-a GV-971 has successfully passed phase III clinical trial in China, and has gradually advanced in the process of drug application, indicating that marine-derived compounds could be used to treat neurodegenerative disorders. Some marine-derived compounds, e.g., **30**–**34** in this review, have entered clinical trials with the aim of treating PD, further providing a support that novel anti-PD drugs might be developed from marine-derived compounds.

With the increasing exploration of the ocean, more marine drugs have been emerged. However, we have only touched the tip of the iceberg for marine resources. Marine organisms could produce a large number of chemicals with novel structures and diverse activities. In addition to compounds that are directly extracted or isolated from marine organisms, designed compounds could be modified and synthesized from the marine-derived leads. 

In this review, 34 marine-derived compounds with pharmacological potential on PD therapy have been summarized, among which 5 compounds have entered into anti-PD clinical trials. Through high-throughput screening and combinatorial chemistry applications, more drug candidates will be found from the marine natural product library for the treatments of PD, especially those could be synthesized directly, or could be obtained in large quantities by fermentation or culture. We anticipated that further focusing on natural products from marine sources may be a promising idea for developing anti-PD leads.

## Figures and Tables

**Figure 1 marinedrugs-17-00221-f001:**
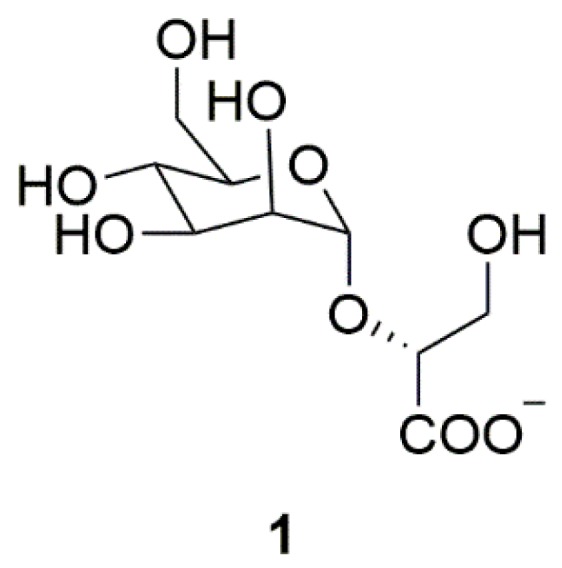
Marine natural product mannosylglycerate (MG) (**1**) derived from archaea.

**Figure 2 marinedrugs-17-00221-f002:**
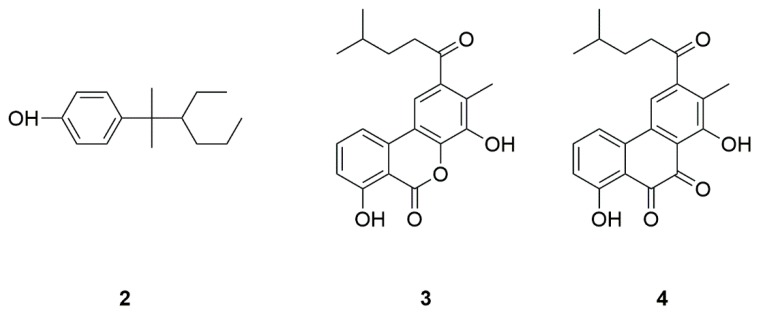
Marine natural products NP7 (**2**), piloquinone A (**3**) and piloquinone B (**4**) derived from bacteria.

**Figure 3 marinedrugs-17-00221-f003:**
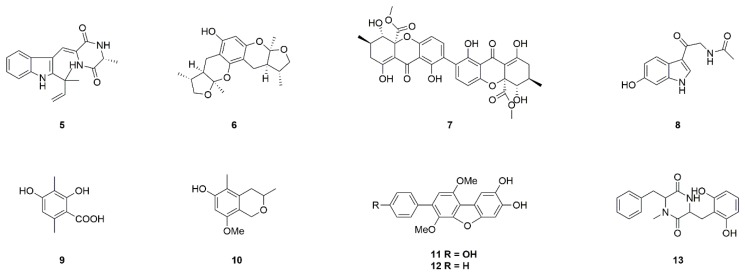
Marine natural products neoechinulin A (**5**), xyloketal B (**6**), secalonic acid A (**7**), 6-hydroxy-*N*-acetyl-β-oxotryptamine (**8**), 3-methylorsellinic acid (**9**) and 8-methoxy-3,5-dimethylisochroman-6-ol (**10**), candidusin A (**11**), 4-dehydroxycandidusin A (**12**) and diketopiperazine mactanamide (**13**) derived from fungi.

**Figure 4 marinedrugs-17-00221-f004:**
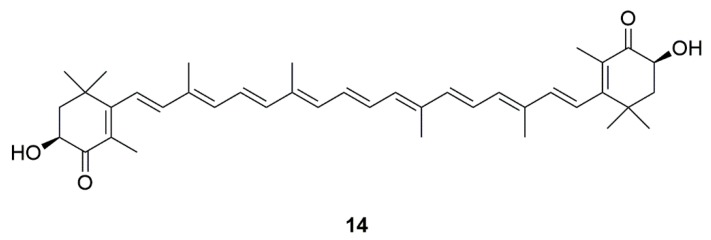
Marine natural product astaxanthin (**14**) derived from algae.

**Figure 5 marinedrugs-17-00221-f005:**
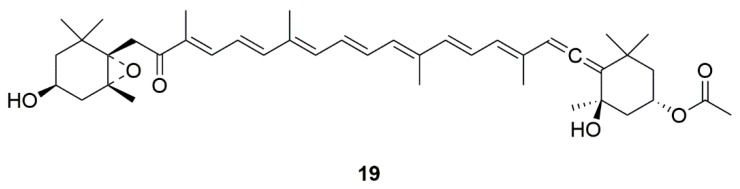
Marine natural product Fucoxanthin (**19**) derived from edible brown algae.

**Figure 6 marinedrugs-17-00221-f006:**
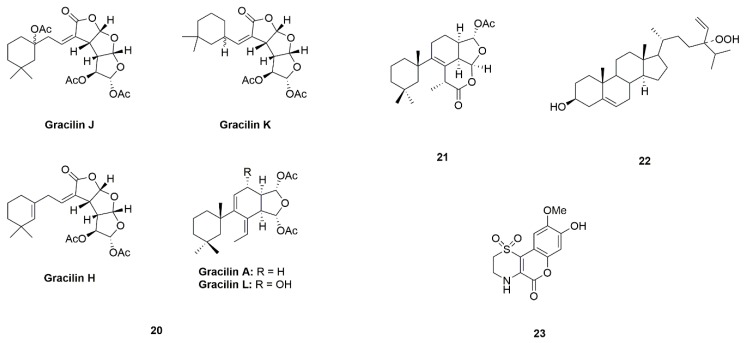
Marine natural products Gracilins (A, H, K, J and L) (**20**), tetrahydroaplysulphurin-1 (**21**), 24-hydroperoxy-24-vinylcholesterol (**22**), and iotrochotazine A (**23**), derived from sponge.

**Figure 7 marinedrugs-17-00221-f007:**
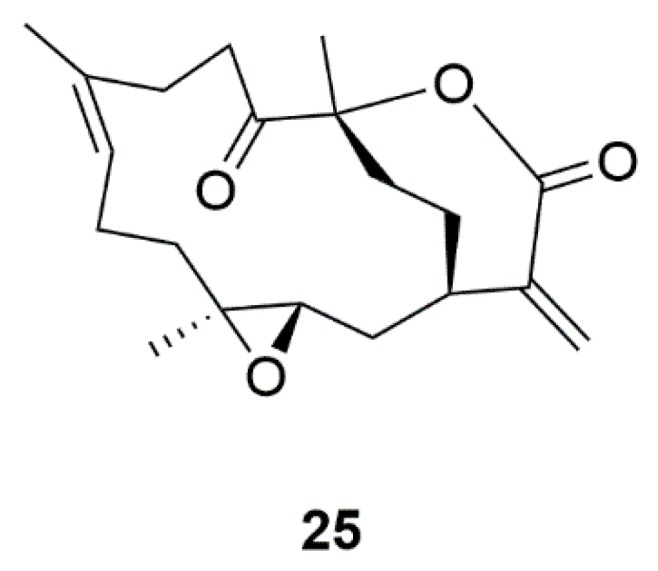
Marine natural product 11-dehydrosinulariolide (**25**) derived from coral.

**Figure 8 marinedrugs-17-00221-f008:**
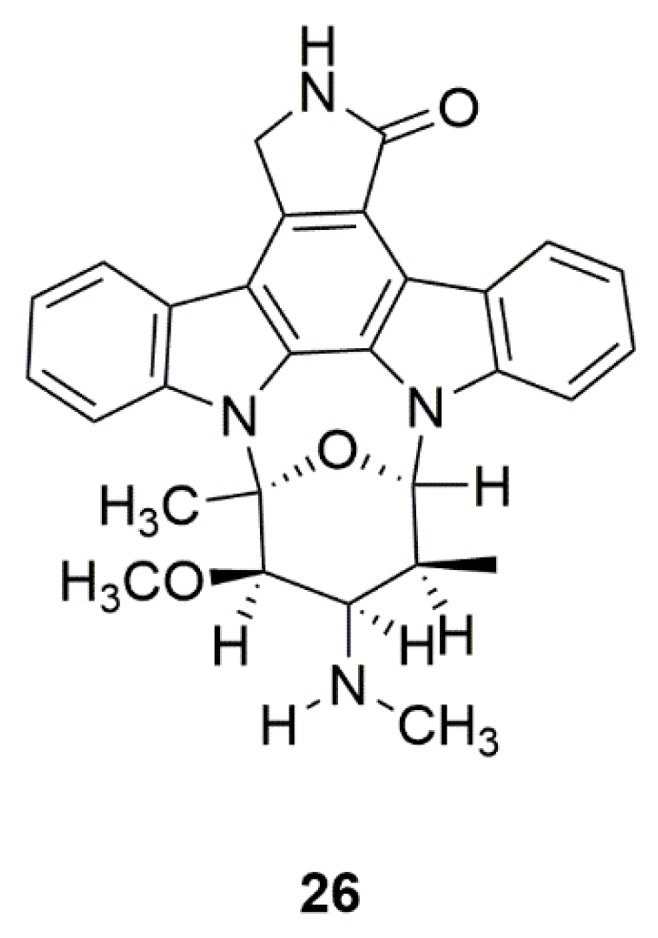
Marine natural product staurosporine (AM-2282) (**26**) derived from prosobranch mollusk, flatworm, and ascidians.

**Figure 9 marinedrugs-17-00221-f009:**
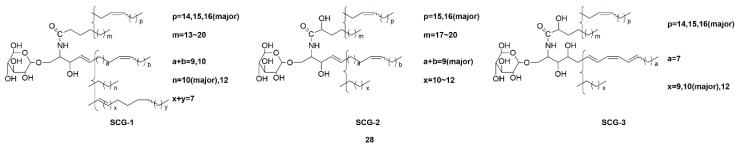
Marine natural product glucocerebrosides (SCG-1, SCG-2, and SCG-3) (**28**) derived from Cucumaria frondosa.

**Figure 10 marinedrugs-17-00221-f010:**
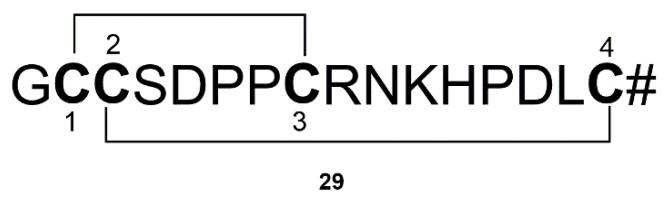
α-Conotoxin (TxIB) (**29**) derived from Conus textile.

**Figure 11 marinedrugs-17-00221-f011:**
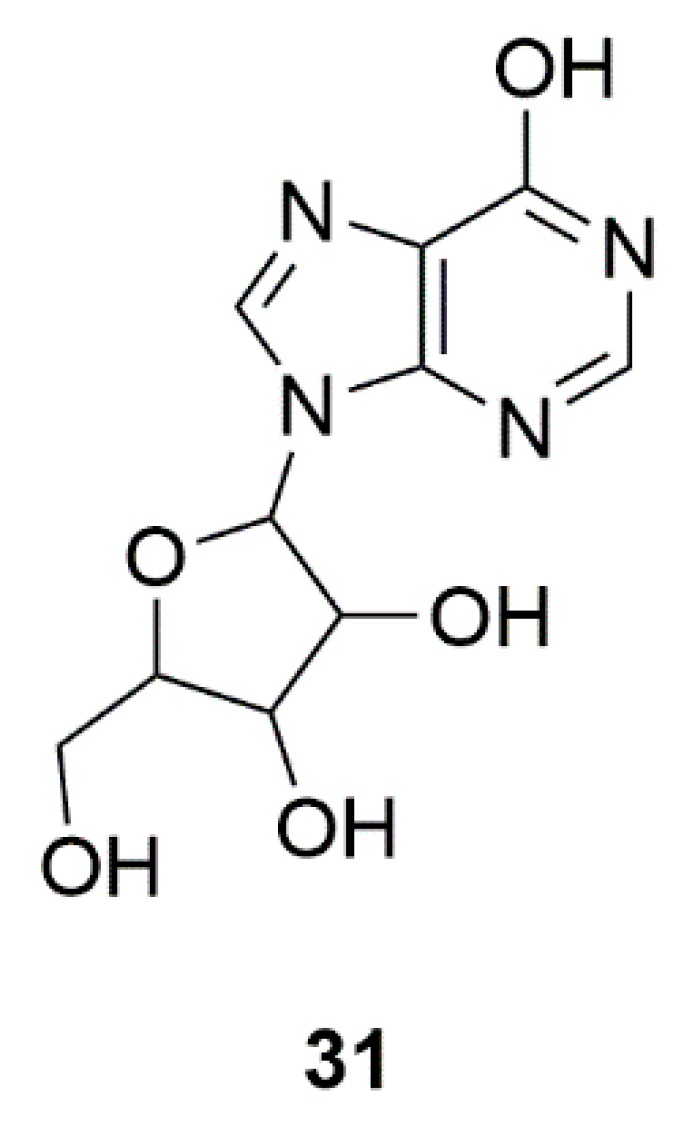
Sponge-derived compound Inosine (**31**).

**Figure 12 marinedrugs-17-00221-f012:**
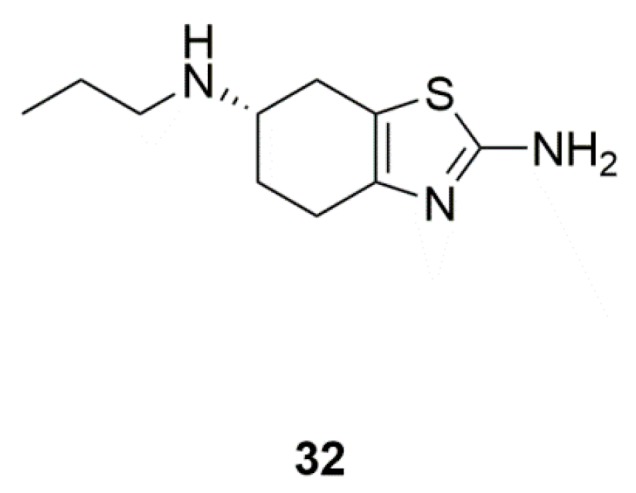
Pramipexole (**32**) could be synthesized with the help of marine yeast.

**Figure 13 marinedrugs-17-00221-f013:**
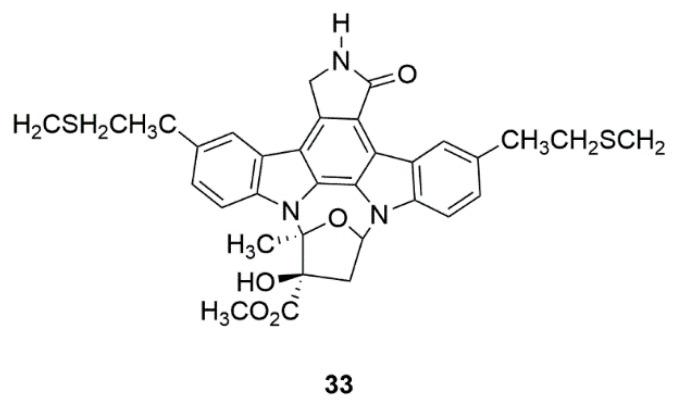
Marine natural product CEP-1347 (**33**) isolated from Nocardiopsis sp. K252a.

**Figure 14 marinedrugs-17-00221-f014:**
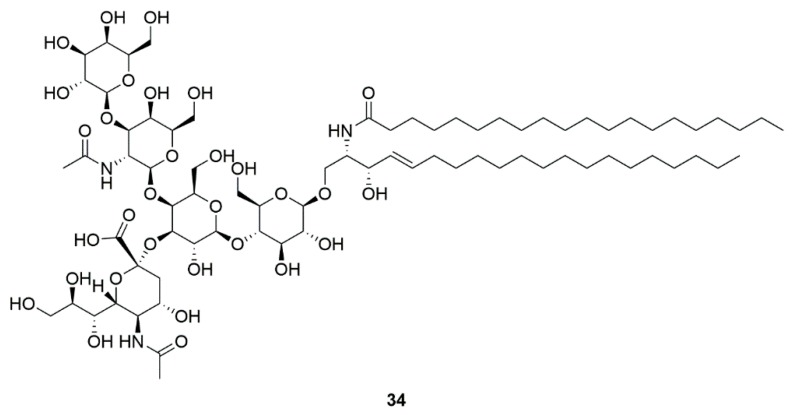
GM1 ganglioside (**34**) can be prepared using a sialidase-producing marine bacterium as a microbial biocatalyst.

**Table 1 marinedrugs-17-00221-t001:** Marine natural products (**1**–**29**) with anti-PD activities, and their mode of actions.

Source	Compounds	Species	Mechanism of Action	Reference
Archaea	Mannosylglycerate (MG) (**1**)	Thermophilic bacteria	Inhibition of α-synuclein aggregation	[11]
Bacteria	NP7 (**2**)	*Streptomyces* sp.	Inhibition of H_2_O_2_-induced neurotoxicity	[12,13,14]
	Piloquinones A (**3**)	Marine-derived *Streptomyces* sp. CNQ-027	Inhibition of MAO-A or MAO-B	[15,16]
	Piloquinones B (**4**)			
Fungi	Neoechinulin A (**5**)	*Microsporum* sp. and *Aspergillus* sp.	Neuroprotection against MPP^+^-induced neurotoxicity	[17,18,19,20]
	Xyloketal B (**6**)	Mangrove fungus *Xylaria* sp. (no. 2508)	Neuroprotection against MPP^+^-induced neurotoxicity	[21,22,23]
	Secalonic acid A (**7**)	*Aspergillus ochraceus* and *Paecilomyces* sp.	Neuroprotection in PD model, inhibition of JNK and p38pathways, Ca^2+^ influx, and caspase-3 activation	[24,25,26]
	6-Hydroxy-*N*-acetyl-β-oxotryptamine (**8**)	*Penicillium* sp. KMM 4672	Protection against 6-OHDA-induced neuronal death	[27,28,29,30,31]
	3-Methylorsellinic acid (**9**)			
	8-Methoxy-3,5-dimethylisocHroman-6-ol (**10**)			
	Candidusin A (**11**)	*Aspergillus* sp. KMM 4676		
	4″-Dehydroxycandidusin A (**12**)			
	Diketopiperazine mactanamide (**13**)	*Aspergillus flocculosus*		
Algae	Astaxanthin (**14**)	*Haematococcus pluvialis* and *Chlorella zofingiensis*	Inhibition of apoptosis, mitochondrial abnormalities, and excessive ROS	[32,33,34,35,36,37]
	Polysaccharide fucoidan (**15**)	*Turbinaria decurrens*	Incensement of antioxidants and dopamine level	[38,39]
	Sulfated hetero-polysaccharides (DF1) (**16**)	*Laminaria japonica*	Activation of the PI3-K/Akt pathway	[40,41,42]
	Sulfated galactofucan polysaccharides (DF2) (**17**)			
	*Spirulina platensis* (**18**)	*Cyanobacterium*	Neuroprotection in α-synuclein-, MPTP-, 6-OHDA-induced models of PD	[43,44,45,46]
	Fucoxanthin (**1****9**)	Edible brown seaweeds	Activation of the PI3-K/Akt cascade and inhibition of the ERK pathway	[47,48]
Sponge	Gracilins (A, H, K, J and L) (**20**)	*Spongionella* sp.	Protection of mitochondrial functions via acting on Nrf2/ARE pathways	[49,50]
	Tetrahydroaplysulphurin-1 (**21**)			
	24-Hydroperoxy-24-vinylcholesterol (**22**)	*Xestospongia testudinaria*	Activation of NF-κB	[51,52]
	Iotrochotazine A (**23**)	*Iotrochota* sp.	Acting on the early endosome and lysosome markers	[53,54,55]
	Mirabamides A–H peptides (**24**)	*Siliquariaspongia mirabilis* and *Stelletta clavosa*	Inhibition of the formation of AGEs	[56,57,58,59]
Coral	11-Dehydrosinulariolide (**25**)	*Sinularia flexibili*	Activation of PI3-K/Akt, p-CREB, and Nrf2/HO-1 pathways	[60,61,62,63]
Mollusk	Staurosporine (AM-2282) (**26**)	Prosobranch mollusk, flatworm, and ascidians	Inhibition of AMPK, and promotion of DA neurite outgrowth	[64,65,66,67]
Sea cucumber	Whole body-ethyl acetate (WBEA), whole body-butanol (WBBU), and body wall-ethyl acetate (BWEA) (**27**)	*Holothuria scabra*	Reduction of α-synuclein aggregation and attenuation of DA degeneration	[68]
	Sea cucumber glucocerebrosides (SCG-1, SCG-2, and SCG-3) (**28**)	*Cucumaria frondosa*	Activation of the TrkA/CREB/BDNF signaling pathway	[69,70]
Conus	α-Conotoxin (TxIB) (**29**)	*Conus textile*	Selectively acting on nAChRs	[71]

**Table 2 marinedrugs-17-00221-t002:** Marine-derived natural products (**30**–**34**) used in clinical trials against PD (Data sources: ClinicalTrials.gov).

Source	Drug	Study Title	Outcome	Reference
Fish and algea	Omega-3 fatty acids (**30**)	Reducing dyskinesia in PD with omega-3 fatty acids (Phase 1); Quality improvement and practice based research in neurology using the EMR (Phase 4)	Improvement of depressive symptoms, decrease of inflammation and oxidative stress	[72,73,74,75]
Sponge	Inosine (**31**)	Safety of urate elevation in PD (Phase 2); Study of urate elevation in PD (Phase 3)	Increase of serum and CSF urate, generally safe and well tolerated	[76,77]
Marine yeasts	Pramipexole (**32**)	Pramipexole versus placebo in PD patients with depressive symptoms (Phase 4)	Direct antidepressant effects	[78,79]
Marine bacteria *Nocardiopsis* sp. (K252a)	CEP-1347 (KT7515) (**33**)	Safety and efficacy study of CEP-1347 in the treatment of PD (Phase 2 and 3)	Identification of serum urate as the first molecular factor directly related to typical PD progression	[80,81,82,83]
Marine bacteria *Pseudomonas* sp.	GM1 ganglioside (**34**)	GM1 ganglioside effects on PD (Phase 2)	Significant improvement in sports score	[84,85,86,87]

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
