# Peer review of "Marine-Derived Natural Compounds for the Treatment of Parkinson’s Disease"

_marinedrugs, 2019, doi:10.3390/md17040221_

Reviewer 1 Report

Manuscript presents an interesting study concernign the merine derived substances to treat Parkinson disease. Currently the topic is of high importance, couse PD seems to be civilization disease. There are some minor language mistakes that should be revised. I would  aslo add to the manuscript some information how these maine-derived natural compounds affect mitochondria and mitochondrial respiratory chain reaction. There would be also interesting the impact on the VDAC1 channel, which is currently one of the therapeutic  targets. 

Author Response

There are some minor language mistakes that should be revised. I would  also add to the manuscript some information how these marine-derived natural compounds affect mitochondria and mitochondrial respiratory chain reaction. There would be also interesting the impact on the VDAC1 channel, which is currently one of the therapeutic targets. 

Response:

Thank you for your suggestion. We have revised the manuscript to improve its English and style. Moreover, we have added the information of VDAC1 channel as below in the manuscript accordingly.

Voltage-dependent anion-selective channel 1 (VDAC1) is located in the outer membrane of mitochondria. ROS produced by mitochondrial complex III could be released into the cytosol from VDAC1 1. In the brain of PD patients, VDAC1 could also increase the concentrations of Ca2+ in the mitochondria, leading to the increase of mitochondrial permeability, the disruption of mitochondrial membrane potential, and eventually neuronal apoptosis and degeneration 2. Therefore, neuroprotective agents that could elevate anti-oxidative stress via acting on VDAC1, ERK, PI3-K/Akt and Nrf2/ARE cascades might produce anti-PD activities.

1. Chen, Q., Vazquez, E. J., Moghaddas, S., Hoppel, C. L., and Lesnefsky, E. J. (2003) Production of reactive oxygen species by mitochondria: central role of complex III, The Journal of biological chemistry 278, 36027-36031.

2. Shoshan-Barmatz, V., Maldonado, E. N., and Krelin, Y. (2017) VDAC1 at the crossroads of cell metabolism, apoptosis and cell stress, Cell Stress 1, 11-36.

Reviewer 2 Report

The Review by Huang et al. describes the marine natural substances which have been shown to have a potential to treat the Parkinson disease. It described the relevant targets in the PD treatment, the anti-PD compounds isolated from the different groups of marine organisms as well as the most promising compounds. The style and English need to be improved.

1) line 58-59 is not correct. There are only 7 marine-derived drugs currently approved for the use in clinic.

2) line 60: “treatment of PD” instead of “treating of PD”.

3) chapters 3 and 4 have the same titles, which are not correct – should be “potential candidate for the treatment of PD”, titles should be different.

4) line 379: “in the land” should be rephrased. “Sea..” instead of “Marine is a treasure…”

Author Response

The style and English need to be improved.

We have revised the manuscript to improve its English and style

1) line 58-59 is not correct. There are only 7 marine-derived drugs currently approved for the use in clinic.

Response:

We have accordingly changed “Many” to “7” accordingly.

2) line 60: “treatment of PD” instead of “treating of PD”.

Response:

We have revised the “treating of PD” to “the treatment of PD” accordingly.

3) chapters 3 and 4 have the same titles, which are not correct – should be “potential candidate for the treatment of PD”, titles should be different.

Response:

We have accordingly revised the “Potential PD candidates from marine-derived compounds” to “Potential candidates from marine-derived compounds for the treatment of PD”. In addition, we have changed chapter 4 to "Marine-derived drugs for clinical trials of PD".

4) line 379: “in the land” should be rephrased. “Sea..” instead of “Marine is a treasure…”

Response:

We have rephrase the sentence as “This is partially because of the lack of finding novel natural compounds from terrestrial organisms .” and used “sea” instead of “marine” in the manuscript.

Reviewer 3 Report

In this review article authors have summarized anti-PD marine-derived natural products which have been shown pharmacological activities via acting on various PD targets, such as α-synuclein, monoamine oxidase B, and reactive oxygen species. Moreover, marine-derived natural compounds currently evaluated in the clinical trials for the treatment of PD are also discussed. The manuscript is very clear and well written, Well done.

Author Response

Thank you for your kindly comments.